



# 1 CASCADE - The Circum-Arctic Sediment CArbon DatabasE

Jannik Martens[1], Evgeny Romankevich[2], Igor Semiletov[3,4,5], Birgit Wild[1], Bart van Dongen[1,6], Jorien
Vonk[1,7], Tommaso Tesi[1,8], Natalia Shakhova[4,9], Oleg V. Dudarev[3], Denis Kosmach[3], Alexander Vetrov[2],
Leopold Lobkovsky[2], Nikolay Belyaev[2], Robie Macdonald[10], Anna J. Pieńkowski[11], Timothy I.
Eglinton[12], Negar Haghipour[12], Salve Dahle[13], Michael L. Carroll[13], Emmelie K.L. Åström[14], Jacqueline
M. Grebmeier[15], Lee W. Cooper[15], Göran Possnert[16], and Örjan Gustafsson[1]
[1]Department of Environmental Science and Bolin Centre for Climate Research, Stockholm University, Sweden
[2]Shirshov Institute of Oceanology, Moscow, Russia
[3]Pacific Oceanological Institute FEB RAS, Vladivostok, Russia
[4]University of Alaska Fairbanks, USA
[5]Northern (Arctic) Federal University, Archangelsk, Russia
[6]Department of Earth and Environmental Sciences and Williamson Research Centre for Molecular Environmental
Science, University of Manchester, United Kingdom
[7]Department of Earth Sciences, Vrije Universiteit Amsterdam, The Netherlands
[8]Institute of Polar Sciences, National Research Council, Bologna, Italy
[9]Scientific Centre Moscow State University (MSU)-Geophysics, Moscow, Russia
[10]Institute of Ocean Sciences, Department of Fisheries and Oceans, Sidney, Canada
[11]Department of Arctic Geology, The University Centre in Svalbard (UNIS), Svalbard, Norway (current address:
Norwegian Polar Institute, Longyearbyen, Svalbard, Norway)
[12]Laboratory of Ion Beam Physics and Geological Institute, ETH Zürich, Switzerland
[13]Akvaplan-niva, FRAM - High North Research Centre for Climate and the Environment, Tromsø, Norway
[14]Department of Arctic and Marine Biology, UiT-The Arctic University of Norway, Tromsø, Norway
[15]Chesapeake Biological Laboratory, University of Maryland Center for Environmental Science, Solomons, USA
[16]Department of Physics and Astronomy, Tandem Laboratory, Uppsala University, Sweden

*Correspondence to*: Örjan Gustafsson (orjan.gustafsson@aces.su.se)





**Abstract**
Biogeochemical cycling in the extensive shelf seas and in the interior basins of the semi-enclosed Arctic Ocean are
strongly influenced by land-ocean transport of carbon and other elements. The Arctic carbon cycle system is also
inherently connected with the climate, and thus vulnerable to environmental and climate changes. Sediments of the
Arctic Ocean are an active and integral part in Arctic biogeochemical cycling, and provide the opportunity to study
present and historical input and fate of organic matter (e.g., through permafrost thawing).
To compare differences between the Arctic regions and to study Arctic biogeochemical budgets, comprehensive
sedimentary records are required. To this end, the Circum-Arctic Sediment CArbon DatabasE (CASCADE) was
established to curate data primarily on concentrations of organic carbon (OC) and OC isotopes ($\delta^{13}$C, $\Delta^{14}$C), yet
also on total N (TN) as well as of terrigenous biomarkers and other sediment geochemical and physical properties
drawn both from the published literature and from earlier unpublished records through an extensive international
community collaboration.
This paper describes the establishment, structure and current status of CASCADE. This first public version includes
OC concentrations in surface sediments at 4244 oceanographic stations including 2317 with TN concentrations,
1555 with $\delta^{13}$C-OC values, 268 with $\Delta^{14}$C-OC values and 653 records with quantified terrigenous biomarkers (high
molecular weight *n*-alkanes, *n*-alkanoic acids and lignin phenols) distributed over the shelves and the central basins
of the Arctic Ocean. CASCADE also includes data from 326 sediment cores, retrieved by shallow box- or multi-
coring and deep gravity/piston coring, as well as sea-bottom drilling. The comprehensive dataset reveals several
large-scale features, including clear differences in both OC content and isotope-based diagnostics of OC sources
between the shelf sea recipients. This indicates, for instance, the release of strongly pre-aged terrigenous OC to the
East Siberian Arctic shelf and younger terrigenous OC to the Kara Sea and thus provides clues about land-ocean
transport of material released by thawing permafrost.
CASCADE enables synoptic analysis of OC in Arctic Ocean sediments and facilitates a wide array of future
empirical and modelling studies of the Arctic carbon cycle. CASCADE is openly and freely available online
(https://doi.org/10.17043/cascade; Martens et al., 2020b), is provided in various machine-readable data formats
(data tables, GIS shapefile, GIS raster), and also provides ways for contributing data for future CASCADE versions.
CASCADE will be continuously updated with newly published and contributed data over the foreseeable future as
part of the database management of the Bolin Centre for Climate Research at Stockholm University.





## 1 Introduction

The Arctic Ocean receives large input of terrestrial organic matter from its large rivers and from coastal erosion, making it both a valuable receptor system for studying large-scale terrestrial carbon remobilization and marine biogeochemistry. Rising temperatures are causing multiple changes to the Arctic, including reduced sea-ice cover, accelerated erosion of ice-rich permafrost shoreline and enhanced river runoff. These changes affect the input of terrestrial organic matter, nutrients and the detrital load, which in turn affects the ocean optical field, marine primary productivity, ocean acidification and many other aspects of biogeochemical cycling in the Arctic Ocean (Stein and Macdonald, 2004; Vonk and Gustafsson, 2013). On land, climate change causes warming and thaw of terrestrial permafrost (Biskaborn et al., 2019), potentially remobilizing parts of its large dormant pool of OC (1300 Pg; Hugelius et al., 2014) into active carbon cycling. By transformation and translocation of previously frozen organic matter, rising temperatures may thus shift balances in the Arctic carbon cycle, leading to system hysteresis effects and translocated carbon-climate feedback (e.g., Vonk and Gustafsson, 2013). Couplings between the large permafrost-carbon pools and amplified climate warming in the Arctic represent a potential "tipping point" in the climate system. These perturbations may affect both OC sequestration in the biosphere and release of climate-forcing greenhouse gases (e.g., AMAP, 2017; IPCC, 2019) as well as other major effects such as the coupling between permafrost carbon remobilization and ocean acidification across the extensive shelf seas (Semiletov et al., 2016).

Continental shelves cover less than 10% of the global ocean area but account for the largest part of OC accumulation in marine sediments and thereby provide an excellent archive for both terrestrial carbon input and marine productivity. The semi-enclosed structure of the Arctic Ocean with its extensive shelves, including the World's largest continental shelf system, the East Siberian Arctic Shelf (ESAS; the Laptev, East Siberian and Russian part of the Chukchi Sea), further accentuate the particular importance of shelf sediments for the carbon cycling in the Arctic. Earlier landmark contributions have provided comprehensive observational perspectives on the distribution of organic matter in marine sediments at the global scale (e.g., Berner, 1982; Romankevich, 1984; Hedges and Keil, 1995). Focusing in greater detail on carbon in the Arctic, the book by Vetrov and Romankevich (2004) "*Carbon Cycle in the Russian Arctic Seas*" and the book edited by Stein and Macdonald (2004) "*The Organic Carbon Cycle in the Arctic Ocean*" provided the first more comprehensive perspectives on the Arctic land-ocean carbon couplings across various regions and synthesized the collected knowledge of carbon sources, transformations and burial in Arctic marginal seas and the central Arctic Ocean. These compilations demonstrated



substantial regional variations in carbon cycling between different Arctic shelf seas, while also acknowledging the
near lack of observational data for key parameters and regions. Since then, substantial progress has been reported
in individual and region-specific studies. Key progress includes advances in isotope and organic geochemistry,
expanding the variety of biogeochemical proxies to trace both sources and organic matter degradation. Stable
carbon isotopes ($\delta^{13}$C-OC) – widely used to distinguish between marine and terrigenous sources in Arctic Ocean
sediments (e.g., Naidu et al., 1993; Mueller-Lupp et al., 2000; Semiletov et al., 2005) – have since then been greatly
supplemented by an expanded use of natural abundance radiocarbon ($^{14}$C-OC). This has not only improved source
apportionment of OC in bulk sediments across Arctic regions and time scales (e.g., Vonk et al., 2012; Goñi et al.,
2013; Martens et al., 2020), but also in sediment density fractions (Tesi et al., 2016b), suspended particulate organic
matter (e.g., Vonk et al., 2010, 2014; Karlsson et al., 2016), and at the molecular level (e.g., Drenzek et al., 2007;
Gustafsson et al., 2011; Feng et al., 2013). Extensive studies of a wide set of molecular biomarkers (e.g., Fahl and
Stein, 1997; Goñi et al., 2000; Belicka et al., 2004; Yunker et al., 2005; van Dongen et al., 2008; Tesi et al., 2014;
Sparkes et al., 2015; Bröder et al., 2016) have provided growing insights in OC distribution and fate, particularly
for terrigenous organic matter. In order to effectively access and interpret the rapidly growing number of
observational data on organic matter in the Arctic Ocean, it would be greatly beneficial to have all these data
organized in a readily-accessible interactive format to facilitate a broad array of wider system assessments and
comparisons.

The overarching objective of this effort is to curate and harmonize all available data on OC in Arctic Ocean
sediments in an open and freely available database. The Circum-Arctic Sediment CArbon DatabasE (CASCADE)
builds on both previously-published and unpublished collections holding information on OC and total N (TN)
concentrations, OC isotopes ($^{13}$C-OC, $^{14}$C-OC) and molecular biomarkers with an initial focus on terrigenous
organic matter (i.e., high molecular weight - HMW $n$-alkanes, $n$-alkanoic acids, lignin phenols) in sediments of all
continental shelves and the deep central basins of the Arctic Ocean. The backbone of CASCADE are large data
collections, including i) OC concentrations, $^{13}$C/$^{14}$C-isotope data and biomarkers from the informal two-decades
long Swedish-Russian collaboration network the International Siberian Shelf Study (ISSS; Semiletov and
Gustafsson, 2009) (e.g., Guo et al., 2004; Semiletov et al., 2005; van Dongen et al., 2008; Vonk et al., 2012; Tesi
et al., 2016a; Bröder et al., 2018; Martens et al., 2019, 2020; Muschitiello et al., 2020); ii) OC concentrations from
the Arctic portion of the "Carbon Database" of the Shirshov Institute of Oceanology, Russian Academy of Sciences
(Romankevich, 1984; Vetrov and Romankevich, 2004); iii) previously-published databases and online collections



(e.g., Pangaea.de) with many contributions from German-Russian partnerships and cruises involving the Alfred-
Wegener-Institute, Germany (e.g., Stein et al., 1994; Mueller-Lupp et al., 2000; Stein and Macdonald, 2004; Xiao
et al., 2015); iv) US and Canadian based research (e.g., Naidu et al., 1993, 2000; Goñi et al., 2000, 2013; Grebmeier
et al., 2006); and v) data from various other contributors that are acknowledged in the database. Furthermore,
CASCADE includes previously unpublished data, some also generated here in the upstart CASCADE effort, to fill
gaps for particularly data-lean regions such as the Barents and Kara Seas, the Canadian Arctic Archipelago, and
the Chukchi Sea.
The aim of the CASCADE effort is to provide a foundation for future studies, including large-scale assessments of
the carbon cycle, such as characteristics of OC input, and its distribution and fate in the Arctic Ocean. This paper
describes the creation and the structure of CASCADE, including a discussion of data availability and quality.

**2 Data collection and methods**
**2.1 The physical compartments: Arctic shelf seas and interior Arctic Ocean basins**
The CASCADE includes OC data from the entire Arctic Ocean with special focus on the seven Arctic continental
shelf seas (Fig. 1; Table 1). Accordingly, a distinction is made among the Central Arctic Ocean and the following
marginal seas: Beaufort Sea, Chukchi Sea, East Siberian Sea, Laptev Sea, Kara Sea, Barents Sea (incl. White Sea),
and the Canadian Arctic Archipelago (we exclude data from Baffin Bay, Foxe Basin and Hudson Bay, as they are
outside the Circum-Arctic scope of the database). For defining the limits of these Arctic shelf seas, Jakobsson
(2002) is followed, which distinguishes the Arctic Ocean constituent seas using hypsometric criteria, defining *shelf*
as the seaward extension of the continental margin until the increase in steepness at the shelf break (Jakobsson,
2002). CASCADE data for the central Arctic Ocean, which was treated as one individual unit, covers all area
beyond the shelf break and includes the continental slope, rise, deep basins and mid-ocean ridges.

**2.2 Georeferencing and sampling**
The coordinate system used for CASCADE is WGS1984 and coordinates are kept in machine-readable decimal
degrees (Latitude in °N, Longitudes in the -180° to 180° format) to harmonize the data across all GIS applications.



The spatial references also include information about the sediment depth interval that the reported data represent
(Table 1). In addition to the geographical coordinates, CASCADE lists the bathymetric water depth of the sampling
point as reported in the primary literature. The core part of CASCADE is in a table format that contains columns
for the station number ('STATION'), geographical coordinates ('LAT'; 'LON'), the name of the expedition and/or
ship ('EXPEDITION'), the year when the sample was taken ('YEAR') and the sediment depth interval
('UPPERDEPTH'; 'LOWERDEPTH'), where the upper depth is equal to 0 cm in the case of surface sediments. In
addition, the table contains a column for water depth ('WATERDEPTH'), all as reported by the data source. For
samples where the sampling year was unknown, the year of the earliest published record was used instead. In cases
where the water depth was not reported, the water depth was estimated using the latest version (v4) of the
bathymetric map of IBCAO (Jakobsson et al., 2020) corresponding to the position of the oceanographic station and
reported in a separate column ('IBCAODEPTH').

**2.3 Surface sediments and sediment cores**
The first stage of the CASCADE development focused on maximizing spatial coverage for surface sediments of
the seven Circum-Arctic shelf sea systems and the central Arctic Ocean. Surface sediments are here defined as
those collected from the water-sediment interface to a depth of maximum 5 cm. Data for surface sediments are
provided in a table ('CASCADEsurfsed_v1.0') as .txt and .xlsx files, as well as a ready-to-use GIS shapefile format.
This database also includes deeper sediments from sediment cores, which represent longer time-scales and add a
third dimension to the geographical referencing. In CASCADE sediment cores are distinguished by three
depositional time scales using the following criteria:

1.   Centennial scale cores (core scale 1) in upper sediments of the Arctic Ocean, e.g., multi corer, Gemini
166        corer, box corer, van Veen grab sampler, other short gravity corers up to 1 m length;
2.   Millennial scale cores (core scale 2) of shelf sediments roughly covering the depositional time frame from
168        the late Holocene to the last glacial/interglacial transition, by e.g., piston corer, long gravity corer, kasten
169        corer;
3.   Glacial cycles scale cores (core scale 3) from the continental slopes or the deeper Arctic Ocean basins
171        covering periods from earlier than the Last Glacial Maximum, including e.g. drill coring on the Circum-
172        Arctic shelves or deep-sea piston cores.






Downcore data are stored in three separate data tables ('CASCADEcorescale1_v1.0';
'CASCADEcorescale2_v1.0'; 'CASCADEcorescale3_v1.0') in addition to the surface sediment files, including a
column for the sampling depth of core subsamples in cm below the sediment surface ('COREDEPTH').

**2.4 Database parameters**
CASCADE contains information, where available, about the concentration as well as isotopic and molecular
composition of OC in marine Arctic sediments. In addition to i) OC concentrations (column 'OC'), the database
includes ii) concentrations of TN ('TN') and iii) the gravimetric ratio of OC/TN ('OC/TN'), which may provide
additional information about the organic matter source (e.g., Goñi et al., 2005; van Dongen et al., 2008).
Furthermore, CASCADE contains data of iv) $\delta^{13}$C-OC ('d13C') as a parameter to distinguish between marine and
terrestrial sources (e.g., Fry and Sherr, 1989), and v) $\Delta^{14}$C-OC ('D14C') to assess the presence of aged organic
matter released from e.g., permafrost deposits (e.g., Gustafsson et al., 2011; Vonk et al., 2012) or from petrogenic
sources such as sedimentary rocks (e.g., Yunker et al., 2005; Goñi et al., 2013) in marine sediments. More details
about the CASCADE parameters and their units are provided in Table 2.

To facilitate further investigations of terrigenous OC input, CASCADE also includes data of terrigenous
biomarkers (Table 2). This first version of CASCADE compiles total concentrations of *n*-alkanes with high
molecular weight (HMW) and $C_{21}$-$C_{31}$ carbon atoms ($\sum C_{21}$-$C_{31}$; column 'HMWALK'), as well as the often
separately-reported more specific *n*-alkanes $\sum C_{27}+C_{29}+C_{31}$ ('HMWALK_SPEC'). CASCADE also contains the
sum of the HMW *n*-alkanoic acids $\sum C_{20}$-$C_{30}$ ('HMWACID'). Both compound classes stem mostly from terrigenous
compartments as they derive from epicuticular leaf waxes of land plants with a typical pattern of dominating odd-
numbered homologues for HMW *n*-alkanes and even-numbered homologues for HMW *n*-alkanoic acids (Eglinton
and Hamilton, 1967). Furthermore, CASCADE holds concentrations of lignin phenols ($\sum$syringyl, vanillyl,
cinnamyl; 'LIGNIN'), which are products from the break-up of the lignin biopolymer, a compound only produced
by vascular plants (Hedges and Mann, 1979). These three compound classes are frequently used as tracers of the
sources and fate of terrestrial organic matter sequestered in Arctic Ocean sediments (Fahl and Stein, 1997; Goñi et
al., 2000; Tesi et al., 2014; Bröder et al., 2016). It is recognized that there are more parameters that could be
included and CASCADE can add further extensions in future versions.




## 2.5 Reference to the original publication

To maintain a high level of transparency, each data source added to CASCADE is fully cited (in the formatting
style of Earth Systems Science Data; ESSD) including a digital object identifier (doi) linked to its reference in the
primary literature next to each parameter column. Accordingly, the CASCADE data sheet distinguishes between a
common reference for OC, TN and OC/TN data ('CN_CITATION') as it is often combined in one measurement,
and separate references for OC isotopes ('d13C_CITATION'; 'D14C_CITATION') and concentrations of
biomarkers ('BM_CITATION'). This facilitates to register multiple measurements for individual oceanographic
stations. A full list of references is separately provided on the CASCADE website and in the Supplementary
Information of this paper.

## 2.6 Data source and quality

A part of CASCADE builds on previous separate and partly inaccessible databases of OC parameters that key
partners of the CASCADE consortium and others have collected over the years. This includes data from the
informal Swedish-Russian led collaboration network called the International Siberian Shelf Study (ISSS; Semiletov
and Gustafsson, 2009) and the "Carbon" database of the Shirshov Institute of Oceanology. This basis for
CASCADE was strengthened by an extensive survey of the peer-reviewed literature and data mining in the grey
literature of scientific cruise reports. All data are fully cited in a separate column (Table 2). We applied a number
of quality criteria and the database records metadata (e.g., sampling technique in the field, sample storage) when
available. The quality criteria for data to be included in CASCADE are:
• Data need to be (geo-)referenceable and located in the target region (i.e. the Arctic Ocean).
• Information about the analysis method is provided by the data source.
• For OC concentrations, values were generated by elemental analyzer (EA) or Rock Eval pyrolysis and
reported as weight-% OC. Total N concentrations and OC/TN ratios are based on EA only.
• All $\delta^{13}$C-OC data stored in CASCADE are based on isotope ratio mass spectrometry (IRMS), often coupled
to an EA and calibrated against the PDB/V-PDB analytical standards.



• For $\Delta^{14}$C-OC the measurements of $^{14}$C data are based on accelerator mass spectrometry (AMS) with $^{14}$C
data reported as $\Delta^{14}$C, fraction modern ($F_m$) or conventional $^{14}$C age in the original publication. We also
kept records of the AMS lab code of the sample if given.
• Terrigenous biomarker analysis was carried out by solvent extraction (for HMW *n*-alkanes and *n*-alkanoic
acids) or by alkaline CuO oxidation of the lignin biopolymer (for lignin phenols) of the sediments, followed
by wet chemistry purification and quantification using gas chromatography analysis with either flame
ionization or mass spectrometry detection.
In addition to the above-mentioned criteria, the aim was to also include information about carbonate removal by
acid treatment prior to the measurement of OC, $\delta^{13}$C-OC and $\Delta^{14}$C-OC. However, details about applied procedures
were missing in most cases and it is therefore assumed that the carbonate fraction was removed from total carbon
prior to OC, $\delta^{13}$C-OC and $\Delta^{14}$C-OC measurements. All meta information (sampling, storage, analysis) to each
CASCADE entry is included in a respective column in the data spreadsheet (Table 2).

**2.7 New gap filling analyses**
**2.7.1 Bulk OC and carbon isotopes**
Gap filling was performed to complement data obtained from literature, cruise reports and in other ways described
above, in regions of particularly poor data density. These efforts thus focused on areas north of western Siberia
(Barents and Kara Sea region) and in the Canadian Arctic Archipelago, using archived sample material that was
provided by CASCADE collaborators. For OC, TN and $\delta^{13}$C-OC analysis, about 10 mg each of a total of 153
freeze-dried sediment samples were weighed in silver capsules and acidified drop-wise with 3 M HCl in order to
remove carbonates. The measurement was carried out using a Carlo Erba NC2500 elemental analyzer coupled to
an isotope-ratio mass spectrometer (Finnigan DeltaV Advantage) in the Department of Geological Sciences,
Stockholm University, with $\pm$3% precision for OC analysis and $\pm$0.15‰ precision for $\delta^{13}$C-OC isotopic
measurements.

Furthermore, a subset of 95 samples was selected for gap-filling bulk-level $^{14}$C-OC analysis at the Tandem
Laboratory, Department of Physics, Uppsala University. A sample amount corresponding to 1 mg OC was weighed
in tin capsules and acidified with 3 M HCl to remove carbonates. Samples with low OC concentrations (<0.5 %)
were placed in small beakers and exposed to acid fumes in a desiccator for 24 h to remove carbonates and



combusted to $CO_2$ in evacuated quartz tubes prior to graphitization at the AMS laboratory. An additional set of 30
gap-filling samples was analyzed for $^{14}$C at the AMS laboratory of ETH Zurich after acid fumigation. The
measurements at Uppsala University had a precision of on average ±1.9% while the precision at ETH Zurich was
on average ±1.1%.
In CASCADE, all new data points are labelled by citing the database ('Martens et al., 2021. CASCADE - The
Circum-Arctic Sediment CArbon DatabasE. Bolin Centre for Climate Research, Stockholm University, Sweden.
doi:10.17043/cascade, 2021.') in the respective reference columns.

**2.7.2 Analysis of lignin phenols**
Gap-filling analysis was also performed for lignin phenols as molecular biomarkers for terrestrial organic matter
using a set of 64 samples from data-lean regions. To extract lignin phenols from marine sediments we applied an
alkaline CuO oxidation protocol using a microwave-based method as originally presented by Goñi and
Montgomery (2000) and followed the same laboratory routine as described in greater detail elsewhere (Tesi et al.,
2014; Martens et al., 2019).

**2.8 Data conversion and harmonization**
Recalculations of literature data (e.g., for unit conversions) were in some cases necessary to harmonize the data to
the standard units as defined in Table 2.
• In CASCADE the concentration of OC is reported in percent (%) of the dry weight; values previously
published as mg OC per g dry weight were divided by a factor of 10.
• CASCADE uses $\Delta^{14}$C with age correction (equation 1) to report the activity of radiocarbon according to
convention (Stuiver and Polach, 1977; Stenström et al., 2011). For radiocarbon values that were reported
as conventional $^{14}$C ages we used equation 2 to calculate the age-corrected $\Delta^{14}$C.

$$\Delta^{14}C = (F_m \cdot e^{\lambda_C(1950 - Y_C)} - 1) \cdot 1000‰ \qquad (1)$$

$$\Delta^{14}C = (e^{-\lambda_L \cdot T_{14C-years}} \cdot e^{\lambda_C(1950 - Y_C)} - 1) \cdot 1000‰ \quad (2)$$

Where $F_m$ is the fraction modern, $\lambda_C$ the decay constant of the Cambridge half-life of $^{14}$C ($T_{1/2\text{-}C}$ = 5730; $\lambda_C$
= 1/8267), $Y_C$ the year of sample collection, $\lambda_L$ the decay constant of the Libby half-life of $^{14}$C ($T_{1/2\text{-}L}$ =
5568; $\lambda_C$ = 1/8033 and $T_{14C\text{-}years}$ the conventional $^{14}$C age.



• All biomarker concentrations of HMW *n*-alkanes and *n*-alkanoic acids are reported as µg per g OC while
lignin phenols are reported as mg per g OC. Biomarker concentrations that in the original publication were
reported as normalized to dry sediment weight were for CASCADE normalized to the OC concentration
of the sample.

## 2.9 Data interpolation

CASCADE provides interpolated GIS raster files (GEOtiff; coordinate system WGS 1984 Arctic Polar
Stereographic) for OC content, $\delta^{13}$C-OC and for $\Delta^{14}$C-OC in surface sediments across the Arctic Ocean. OC data
was mapped in ArcGIS 10.6 and interpolated to a resolution of 5x5 km per grid cell using the Empirical Bayesian
Kriging function (EBK; Gribov and Krivoruchko, 2020) in the commercially-available ArcGIS 10.8 software
package (ESRI). Kriging builds on the assumption that two points located in proximity are more similar than two
points further distant and creates a gridded surface of predicted values using an empirical semivariogram model.
As an advancement to Kriging, EBK repeatedly simulates semivariogram models in subsets of up to 200 data points
and thus not only improves the prediction but also optimizes interpolation across areas with strongly varying data
availability in the Arctic Ocean (e.g., shelf seas vs. central basins).

## 3 Results and Discussion

## 3.1 Data set inventory

For surface sediments, CASCADE includes 4244 different locations across the Arctic Ocean (Fig. 2), for which
the OC concentration is known, while the concentration of TN, including the OC/TN ratio, is known for 2317
locations (Table 1). For carbon isotopes, the number of individual $\delta^{13}$C-OC values is 1555 and for $\Delta^{14}$C-OC it is
268. CASCADE also holds concentrations of terrigenous biomarkers at 131-213 locations per compound group, of
which most are for HMW *n*-alkanes, either concentrations of HMW *n*-alkanes ($\sum$C$_{21}$-C$_{31}$; 213 stations) or of *n*-
alkane chain lengths more specific for higher plants ($\sum$C$_{27}$, C$_{29}$, C$_{31}$; 164). Fewer data are available for
concentrations of HMW *n*-alkanoic acids ($\sum$C$_{20}$-C$_{30}$; 131) and the concentrations of lignin phenols (145).





In addition to surface sediments, a total number of 326 sediment cores across the Arctic Ocean is included in this
first version of CASCADE. Combined, these hold another 10553 observations of OC concentrations, 4769
concentrations of TN and 2122 $\delta^{13}$C-OC ratios.

## 3.2 Spatial distribution of data

The data coverage for surface sediments is highly variable among the shelf seas, yet improved by the extensive
gap-filling analysis (Table 1). The largest number of OC concentrations is in the Barents Sea (1092; Table 1).
Despite the large total number of available Arctic sediment OC concentrations, there are only 236 samples analyzed
for $\delta^{13}$C-OC and 33 with $\Delta^{14}$C-OC in the Barents Sea, and of these most are located in the Norwegian (western)
sector of the Barents Sea. For the eastern Siberian Arctic and the North American sector of the Arctic Ocean,
observations of OC concentrations are lower but the availability of $\delta^{13}$C-OC data is higher (Table 1, Fig 2b, c).
Accordingly, the Kara, Laptev, East Siberian and Chukchi Seas each support more than 200 $\delta^{13}$C-OC observations.
The number of $\Delta^{14}$C-OC observations is generally lower but reveals highest coverage in near-coastal areas, with
28 values in the Kara Sea, 42 values in the Laptev and 71 values in the East Siberian Sea. Data availability in the
Chukchi Sea for $\Delta^{14}$C-OC is lower (n=12), stressing the need for future analysis. The lowest availability of data is
in the Canadian Arctic Archipelago where the gap-filling analysis of OC, as part of this study, increased the number
of OC concentrations from 21 to 54, with a similarly low number of carbon isotopes (51 of $\delta^{13}$C-OC; 22 of $\Delta^{14}$C-
OC) distributed over its vast area of 1,171,000 km$^2$. The largest individual regime area is covered by the interior
basins of the Central Arctic Ocean, which holds 529 observations of OC concentrations, 130 of $\delta^{13}$C-OC and 27 of
$\Delta^{14}$C-OC values.

## 3.3 Assessment of data quality

Most sources of data used to populate CASCADE provided detailed information about the techniques involved in
analyzing OC concentrations, isotopes and biomarkers, or cited references or cruise reports that contained this
information. The development of CASCADE included the collection of meta information about sampling, storage
and analysis, as described in section 2.6. This information is included and detailed in CASCADE. The quality
assurance information shows that 86% of the reported OC concentrations were analyzed using EA and only a
minority was analyzed by Rock Eval pyrolysis. For $\delta^{13}$C-OC, in 66% of the cases IRMS coupled to EA was given





as the method of analysis. Regarding sample storage, information was given in about 59% of all data sources that
the samples were kept frozen between sampling and analysis, while for <1% of the cases it was documented that
the samples were stored refrigerated; this means that for 40% of the samples, there was no information provided
about sample storage. For 78% of the $\Delta^{14}$C-OC values, the laboratory AMS label was documented and thus also
added to the CASCADE sheet.

**3.4 Circum-Arctic carbon features**
Visualization of CASCADE data directly reveals several large-scale features of OC in Arctic Ocean sediments.
These include clear differences in both OC concentration and source-diagnostic isotope composition among the
shelf seas. For instance, interpolated OC concentrations (Fig. 2) indicate that high sedimentary OC content is found
both in regions of high terrestrial input (e.g., Kara Sea, Laptev Sea, East Siberian Sea and Beaufort Sea) and in
regions of high nutrient availability and marine primary productivity (Barents Sea and Chukchi Sea). The
combination of $\delta^{13}$C and $\Delta^{14}$C isotope values delineate large-scale differences in OC sources. Values of $\delta^{13}$C-OC
close to marine OC (-21‰; Fry and Sherr, 1989) and $\Delta^{14}$C reflecting contemporary carbon are consistent with high
marine primary productivity in the Barents Sea and Chukchi Sea. By contrast, the Kara Sea, receiving input from
major West Siberian catchments (Ob and Yenisey rivers), appears to reflect OC from contemporary terrestrial
sources (~-27‰; Fry and Sherr, 1989), while the terrigenous OC fraction in the Laptev and East Siberian Seas is
much older with a presumably substantial contribution from remobilization of thawing permafrost or other old
deposits via erosional or fluvial processes (Fig. 1; Fig. 2). These and other features can now be investigated through
CASCADE at greater quantitative detail over large intra- and inter-system scales.

**4 Vision and future development**
The CASCADE is the largest and most comprehensive open-access database of OC parameters for Arctic Ocean
sediments. It is a resource that can facilitate a wide range of investigations on OC cycling in the high northern
latitudes, which, for instance, may address research questions on sources of organic matter, marine primary
production, OC degradation, OC transport both in the offshore direction and vertically from the sea surface to the
sediment; and all this both in the contemporary and the historical perspectives. CASCADE provides opportunities



to expand our still limited understanding of how sensitive terrestrial permafrost in different circum-Arctic regions
is towards remobilization both in the current and over earlier periods of rapid climate change.

## 5 Data availability

CASCADE will be hosted and actively updated and extended by a database management at the Bolin Centre for
Climate Research at Stockholm University. CASCADE is accessible at the Bolin Centre data repository
(https://doi.org/10.17043/cascade; Martens et al., 2020b). When using the CASCADE, this paper and the database
should be cited. The website also includes contact details, which can be used to submit new data for incorporation
into future versions of CASCADE – a community effort and resource.





**Author contributions**

The CASCADE database was conceptualized and planned by a team led by ÖG, IS and ER. JM, NB, BW and ÖG developed the technical framework of the CASCADE. JM executed the development of the CASCADE, populated the database with published and unpublished data from the literature and internal records, coordinated gap-filling analyses and created maps. JM drafted and coordinated the manuscript in close collaboration with ÖG and BW. All authors contributed to the realization of the CASCADE database and participated in the editing of the manuscript.

**Acknowledgements**

We thank collaborators throughout and beyond the International Siberian Shelf Study (ISSS) network, and all participants of the Arctic Partner Forum 2018 for their advice in constructing the CASCADE database and for pointing out data sources during the development of the database. We also thank the crew and the scientific party of the ISSS-08 expedition onboard *r/v* Yacob Smirnitskyi, the SWERUS-C3 expedition onboard the *i/b* Oden in 2014 and various other field campaigns organized by the ISSS in 2004, 2005, 2007, 2008, 2011, 2016 and 2017. Furthermore, Lisa Bröder and Rickard Wennström are thanked for their help with gap-filling analysis of lignin phenols. We also thank August Andersson and Henry Holmstrand for their long-term assistance and advice during various field campaigns, laboratory analyses and computer-based work that contributed to the realization of this database.

**Funding**

Development of the CASCADE was supported by the European Research Council (ERC Advanced Grant CC-TOP 695331 to ÖG), the EU H2020-funded project Nunataryuk (Grant 773421) and the Swedish Research Council (Grant 2017-01601). Field campaigns to obtain gap-filling samples were supported by the Knut and Alice Wallenberg Foundation (KAW contract 2011.0027 to ÖG) as part of the SWERUS-C3 program, as well as by the Russian Government (grant 14, Z50.31.0012 to IS) and the Russian Science Foundation (grant 15-17-20032 to NS). Furthermore, this study was supported by the assignment of the Russian Academy of Sciences (grant 0149-2019-0006) and the Russian Science Foundation (grant 18-05-60214) to the Shirshov Institute of Oceanology (ER, AV). The collection of sample material in the Barents Sea was supported by the Research Council of Norway (grant 228107 to MLC; grant 223259) and VISTA (grant 6172 to EKLÅ). Gap-filling samples from the Canadian Arctic



were supported by the Research Council of Canada (NSERC Discovery Grant RGPIN-2016-05457 to AJP). BvD
was supported by NERC research grant (NE/I024798/1) and JV was supported by the Dutch-NWO (Veni grant #

403  863.12.004).


**Competing interests**
The authors declare that they have no conflict of interest.



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





**Figure 1:** Overview map of the Arctic Ocean compartments defined and used in CASCADE, with the permafrost
distribution based on numerical modelling (Obu et al., 2019), rates of coastal erosion (Lantuit et al., 2012); and the
latest IBCAO v4 bathymetry (Jakobsson et al., 2020). Black lines delineate the extent of the Arctic Ocean shelf
seas and each respective watershed on land.



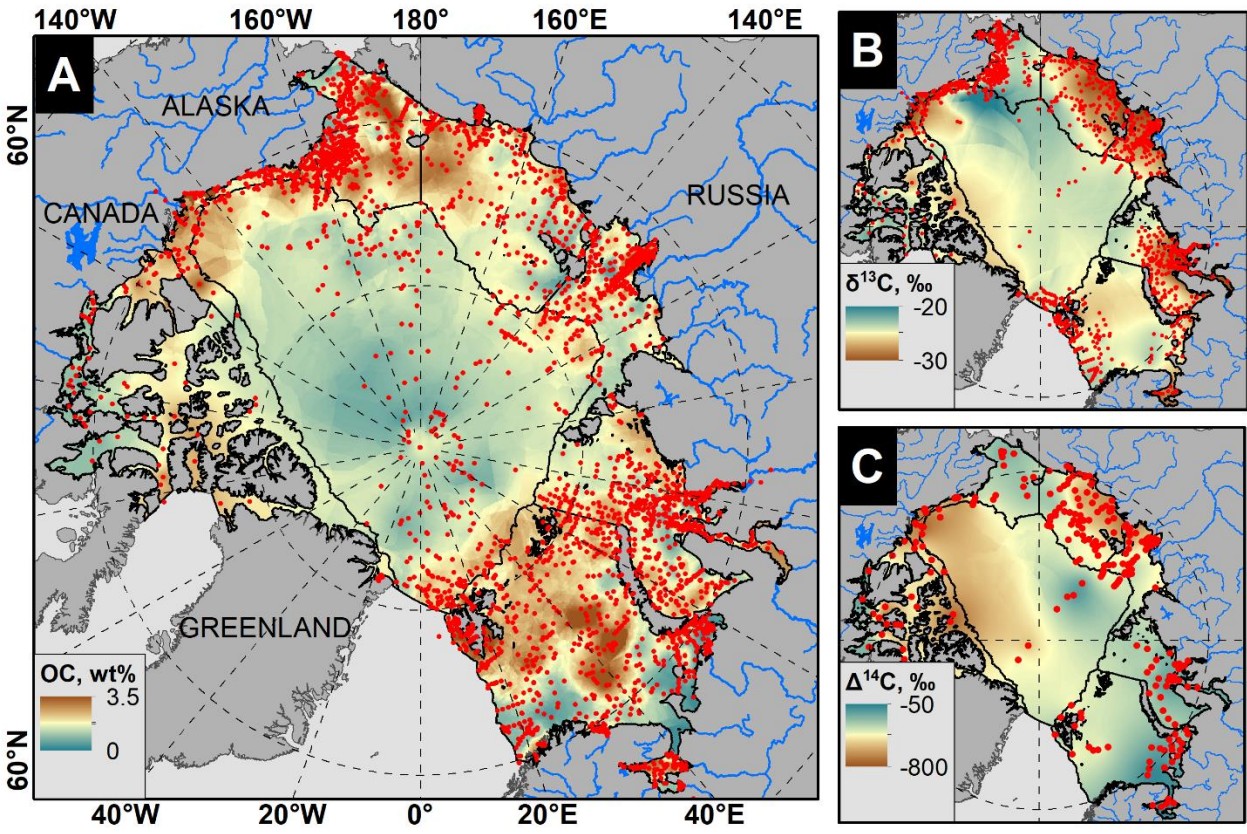

**Figure 2:** CASCADE data location for OC concentrations (panel A), carbon isotopes $\delta^{13}C$ (panel B) and $\Delta^{14}C$
(panel C) marked as red dots, with interpolated fields as indicated by the inserted color scale and as described in
the main text.



**Table 1: CASCADE data availability per circum-Arctic shelf sea and for the interior basin**

| Shelf area | Area $10^3$ km$^2$ $n$ | OC $n$ | TN $n$ | $\delta^{13}C$ $n$ | New $\delta^{13}C$ $n$ | $\Delta^{14}C$ $n$ | New $\Delta^{14}C$ $n$ | Alk1[1] $n$ | Alk2[2] $n$ | Acid[3] $n$ | Lignin[4] $n$ | New lignin $n$ |
|---|---|---|---|---|---|---|---|---|---|---|---|---|
| 1. Barents Sea[5] | 1626 | 1092 | 353 | 236 | 48 | 33 | 33 | 0 | 13 | 0 | 0 | 0 |
| 2. Kara Sea | 942 | 637 | 201 | 262 | 22 | 29 | 22 | 2 | 90 | 2 | 0 | 0 |
| 3. Laptev Sea | 505 | 312 | 110 | 214 | 8 | 42 | 14 | 33 | 46 | 31 | 36 | 19 |
| 4. East Siberian Sea | 1000 | 259 | 217 | 187 | 17 | 71 | 16 | 28 | 13 | 10 | 68 | 40 |
| 5. Chukchi Sea | 639 | 1084 | 950 | 256 | 9 | 12 | 10 | 67 | 14 | 58 | 3 | 0 |
| 6. Beaufort Sea | 183 | 247 | 122 | 219 | 5 | 32 | 3 | 5 | 1 | 2 | 11 | 0 |
| 7. Canadian Arctic Archipelago[6] | 1171 | 92 | 87 | 55 | 29 | 22 | 19 | 0 | 0 | 0 | 9 | 0 |
| 8. Central Arctic Ocean[7] | 4500 | 529 | 282 | 130 | 15 | 27 | 10 | 29 | 36 | 28 | 18 | 5 |
| Total | 10566 | 4252 | 2322 | 1559 | 153 | 268 | 127 | 164 | 213 | 131 | 145 | 64 |

[1] Alk1: HMW $n$-alkanes $\Sigma C_{21}$-$C_{31}$
[2] Alk2: HMW $n$-alkanes $C_{27}$+$C_{29}$+$C_{31}$
[3] Acid: HMW $n$-alkanoic acids $\Sigma C_{20}$-$C_{30}$
[4] Lignin: lignin phenols syringyl, vanillyl and cinnamyl
[5] incl. White sea and shelf northwest of Svalbard
[6] incl. shelf northeast of Greenland
[7] incl. continental slope, rise and abyssal plain



**Table 2: Parameter description and name of the respective columns in the CASCADE data sheet**

| Parameters | Description | Column name |
|---|---|---|
| CASCADE entry ID | Serial number | ID |
| *Georeference and sampling information* | | |
| Sample code | Expedition station ID | STATION |
| Latitude | Decimal latitude according to WGS1984 | LAT |
| Longitude | Decimal longitude according to WGS1984 | LON |
| Upper sample depth (cm) | Sample depth (for surface sediments only) | UPPERDEPTH |
| Lower sample depth (cm) | Sample depth (for surface sediments only) | LOWERDEPTH |
| Median sample depth (cm) | Median sample depth (for core samples only) | COREDEPTH |
| Water depth (m b.s.l.) | Water depth of sampling according to shipboard measurement | WATERDEPTH |
| Water depth based on IBCAO (m b.s.l.) | Water depth according to IBCAOv4 | IBCAODEPTH |
| Expedition or vessel name | Vessel name, expedition name, cruise number | EXPEDITION |
| Sampling year | Year when the sample was taken as reported in literature | YEAR |
| *Carbon and Nitrogen (CN) data* | | |
| OC (%) | Total OC concentration of the bulk sediment; carbonate removal assumed | OC |
| TN (%) | Total N concentration of the bulk sediment | TN |
| OC/TN | OC/TN ratio (gravimetric); published values or calculated | OC_TN |
| *Carbon isotopes* | | |
| $\delta^{13}$C (‰ VPDB) | $\delta^{13}$C-OC; carbonate removal assumed | d13C |
| $\Delta^{14}$C (‰) | $\Delta^{14}$C-OC corrected for age; carbonate removal assumed | D14C |
| *Biomarkers* | | |
| *n*-alkanes $C_{21-31}$ (µg g$^{-1}$ OC) | OC-normalized concentration of HMW *n*-alkanes | HMWALK |
| *n*-alkanes $C_{27,29,31}$ (µg g$^{-1}$ OC) | OC-normalized concentration of specific HMW *n*-alkanes | HMWALK_SPEC |
| *n*-alkanoic acids $C_{20-30}$ (µg g$^{-1}$ OC) | OC-normalized concentration of HMW *n*-alkanoic acids | HMWACID |
| lignin phenols (mg g$^{-1}$ OC) | OC-normalized concentration of syringyl, vanillyl, cinnamyl | LIGNIN |
| *Quality parameter and meta information* | | |
| Sediment sampler | Method of sediment sampling | SAMPLER |
| Sample storage | 0 unknown, 1 frozen, 2 refrigerated, 3 dried onboard | STORAGE |
| CN measurement | Description of the method of analysis of the OC and TN data | CN_METHOD |
| $^{13}$C measurement | Description of the method of analysis of $^{13}$C-OC | d13C_METHOD |



| $^{14}$C measurement | Description of the method of analysis of $^{14}$C-OC | D14C_METHOD |
| AMS label | Laboratory number of the $^{14}$C measurement | D14C_LABEL |
| | | |
| *Citation of the data source* | | |
| Citation of CN data | Full citation in ESSD style incl. info about publication format | CN_CITATION |
| Citation of $\delta^{13}$C data | Authors, title, journal, volume, pages, doi, year | d13C_CITATION |
| Citation of $\Delta^{14}$C data | Full citation in ESSD style incl. info about publication format | D14C_CITATION |
| Citation of biomarker data | Full citation in ESSD style incl. info about publication format | BM_CITATION |
