# Peer review of "CASCADE - The Circum-Arctic Sediment CArbon DatabasE"

_Earth System Science Data, 2020_

## Referee Comment (RC1) · Gerrit Müller (Referee) · 27 Jan 2021

General Comment

The authors generated a comprehensible, freely accessible dataset on organic carbon concentration, its isotopic composition, nitrogen concentration and terrigenous biomarkers in circum-Arctic marine sediments. This is an original and useful compilation, as it makes data published in different sources easily accessible, previously unpublished data available and even complements existing data with gap-filling measurements and spatial interpolation using GIS. Such data is needed for integrative and large-scale assessment of biogeochemical cycles, especially in regions as sensitive to changing climate as the Arctic. First insights are deduced and visualized, identifying

spatial variation in organic carbon concentration and sources, which gives the reader an impression of how to use and interpret this data. The provided dataset is well organized, curated, (re)traceable and citable via DOI, despite a few apparent transcription errors (which are common in these types of manual compilations) and minor comments on comprehensibility (see Dataset comments). The paper itself describes data derivation and classification in a reasonable way and with sufficient detail, although (apart from minor comments) the reader-friendliness of the structure, consistency and conciseness could be improved.

Specific Comments

Abstract The Abstract includes all relevant information, but could be more concise. For instance, the beginning (L.30 – 34) can be reformulated more concise and your interpretation presented in L. 46 – 50 may be explained more briefly. Possibly, try to avoid too many adjectives/adverbs and embedded subordinate clauses.

Introduction 1.) In L. 58: '[..] large input of terrestrial organic matter from its large rivers and from coastal erosion, making it both a valuable receptor system for studying large-scale terrestrial carbon remobilization and marine biogeochemistry', consider substitution of one or two 'large'. There is also a semantic error: '[. . .] making it both a valuable receptor system [. . .] and marine biogeochemistry.' You may try to relocate 'both'. 2.) References are needed to support your statement, that warming accelerates coastal erosion and river runoff in L. 60 - 61 (although this is senseful of course). Similarly, citations should be added in L. 69-70 (arctic warming as a tipping point in the climate system) and L. 75 – 79 (global and arctic shelf area portions). In L. 89 – 90, where you mention 'Key progress', the (or some) relevant articles should be cited. If that refers to the references in the following explanatory sentences, please make that more clear, e.g. by inserting ':' . 3.) In L. 108 – 109, you state that there is an initial focus on terrigenous organic matter, but not why. Is that because of data availability or because of the database applications you had in mind? Please give the reasoning here.

Data collection and methods 1.) Section2.2 Georeferencing and sampling is a bit confusing (e.g., what do you mean by 'core part'?), which is clarified later in section 2.4 Database parameters. So maybe consider explaining the parameters and structure before you give details on how these parameters are acquired. 2.) L. 151 states that you used '[. . .] the year of the earliest published record [. . .]' when the sampling date was not available. This needs to be visible in the data tables, for full comprehensibility and because publication and sampling year may differ in fact. Is more detailed sampling time information available? Seasonality may play a role in surface sediments. 3.) In section 2.3 you could possibly provide the number of sediment cores available for each of the scales (Centennial, Millennial and Glacial cycles scale) and provide a reasoning for this separation (i.e., how did you choose length range and corresponding time-scale?) 4.) In L. 206 – 209, it is stated that for some samples, variables are from different references. Is that exactly the same sample (or a split)? How do you assure this? Is it explicitly stated by the references? 5.) The detailed description of the included parameters in section 2.4 Database parameters is (partially) repeated in section 3.1 Data set inventory, where you basically make a similar detailed description, but with numbers of samples. 6.) Section 2.6 Data source and quality. In this section you describe 'The quality criteria for data to be included [. . .]'. These appear well chosen at the first glance, but later in the evaluation of the data quality, it becomes obvious that not all data fulfill these criteria (e.g. 'For áž§13C-OC, in 66 % of the cases IRMS coupled to EA was given [. . .]' (L. 333 -334) and the rest?). Therefore, you may not call these selection criteria, but quality assurance criteria or similar. At least stay consistent with this. 7.) In section 2.7.1, L. 259 – 260, you give the precision of the gap-filling measurements, but neither mention how this was acquired (multiple measurements?), nor include a statement regarding accuracy. 8.) The constants used for conversion of 14C data should be referenced (L. 282 – 284).

Results and Discussion 1.) Section 3.1: Partial repetition of section 2.4. See comments in previous section. 2.) Section 3.3: Does your remark in L. 332-334, '[. . .] CASCADE provided detailed information [. . .] or cited references or cruises
that contained this information', mean that it may be up to the user to look up methods for some samples in the original reference? If yes, why didn't you include this information for all samples then? If no, please try to reformulate to avoid confusion. 3.) Which external data sources are available that can be interoperated with CASCADE? It may be instructive to compare marine sediment data to river input, water column chemistry, marine production, coastal and soil erosion or the non-organic carbon part of marine sediment. At least for some of these parameters, databases exist (e.g. GLORICH https://doi.org/10.1594/PANGAEA.902360 or db seadbeds https://instaar.colorado.edu/~jenkinsc/dbseabed/). You may stimulate users by providing some advice here. 4.) You may possibly want to insert a disclaimer in the end (https://www.earth-system-science-data.net/submission.html#manuscriptcomposition).

Dataset 1.) CASCADE is very well structured, comprehensible and user friendly. However, an indication for what type of time-reference is given should accompany that valuable information. In the current version, a user cannot know if a sampling time refers to the real sampling year or to the earliest publication year from the same sample/core, which may differ by some years. 2.) The dataset has proven robust against detailed inspections and logical tests. In spite, there is a little transcription (?) error, identified by the histogram of '% OC' in 'CASCADEcoresv1', indicating an impossible organic carbon concentration of $\sim$ 120 %, which probably propagated into the following calculations (C/N ratios and normalization to organic carbon concentration). Please check your calculations, possible errors in unit conversion and transcription errors (digit typo?) and correct this. 3.) A table translating abbreviations used in the database (e.g. IRMS) and units should accompany the data for clarity and convenience, despite this information is included in the article. 4.) If possible, please provide the interpolated grid also as datafile to complement the rather impractical tiff-file.

---

## Author Comment (AC1) · 29 Jan 2021

We gratefully thank the reviewer for the well-informed and constructive comments to our manuscript, which we will address in detail in our final response and revised manuscript once the public discussion is closed. We specifically appreciate the reviewer for pointing out an error in one of the data sheets in CASCADE. The value for total OC in row ID 6045 in data table 'CASCADEcores_v1.0' and in row ID 5555 in table 'CASCADEcorescale2_v1.0' (133 wt% OC), imported from another data source, is indeed incorrect. The value will be corrected in our revised version of CASCADE, along with other improvements to the dataset and manuscript as suggested by this review.

[Figure]

2020.

---

## Referee Comment (RC2) · Anonymous Referee #2 · 22 Mar 2021

General Comments: Martens et al. present a new, openly accessible database called "CASCADE" compiling various data from surface sediments and sediment cores from the Arctic Ocean. Specifically, the database contains data on TN and TOC contents, carbon isotopic compositions of bulk organic matter ($\delta$13C and $\Delta$14C) and concentrations of terrigenous biomarkers (n-alkanes, n-alkanoic acids and lignin phenols). The authors combine existing data from different databases and even make previously inaccessible data available for the community. They also generate new data to fill regional gaps. At the end of the paper, they apply CASCADE to interpolate carbon concentrations and bulk 13C and bulk 14C data over the Arctic Ocean and discuss regional differences.

The paper and the database are clearly arranged and comprehensible. The data is easy to access as it is provided in common formats including text, excel and shape files. The quality standards and methods applied are state of the art and robust.

The database is a valuable piece of work and will be vital for Arctic research as it allows to comprehensively study biogeochemical processes and carbon cycling in the Arctic Ocean and how these processes are coupled to changes in the adjacent continents (e.g. permafrost thawing). Understanding these processes is crucial in a region expected to change drastically in the course of global warming. Altogether the database and paper stand already very well by themselves and do not need major editing. The paper can be published soon after a few small issues are addressed.

1) Overall the paper is well structured and easy to understand. However, it contains some rather long sentences that should be shortened to facilitate reading. Moreover, I found a few instances where the word order should be rearranged (see Specific Comments below) 2) Although the authors describe the file contents on the website and in the paper, I find it unhandy that the data files do not contain units in the column headers. The authors should consider to add the units to the files to make the work with the data even more comfortable for the user.

Specific comments: Line 43: change "shelves" to "Shelves"

Line 111: I think "13C/14C-isotope data" actually means $\delta$13C and $\Delta$14C data. This should be clearly stated as the term may be misunderstood as the isotopic ration between 13C and 14C.

Line 158: change to: "Here, surface sediments are defined as. . ."

Section 2.7.1: Does the gap filling concern surface sediments and cores?

Line 366: change to: towards remobilization in both, the current and over earlier. . ..

---

## Author Response (AR2)

**Author response to reviews to ESSD to ms essd-2020-401 "CASCADE – The Circum-Arctic Sediment CArbon DatabasE"**

Ref: ms. no. essd-2020-401

Jannik Martens, Evgeny Romankevich, Igor Semiletov, Birgit Wild, Bart van Dongen, Jorien Vonk, Tommaso Tesi, Natalia Shakhova, Oleg V. Dudarev, Denis Kosmach, Alexander Vetrov, Leopold Lobkovsky, Nikolay Belyaev, Robie Macdonald, Anna J. Pieńkowski, Timothy I. Eglinton, Negar Haghipour, Salve Dahle, Michael L. Carroll, Emmelie K.L. Åström, Jacqueline M. Grebmeier, Lee W. Cooper, Göran Possnert, and Örjan Gustafsson

> We gratefully thank the reviewers for constructive comments that have clearly contributed to improve the manuscript and the CASCADE during revision. We are encouraged by the reviewers' overall positive assessments of the manuscript and the database. All reviewer comments and our responses are listed below, organized such that the reviewer comments are shown first in *italics black font*, followed by our response in normal blue tab-indented text. Our response refers to line numbers in the revised manuscript version.

**Referee comment 1**

**General Comment**

*The authors generated a comprehensible, freely accessible dataset on organic carbon concentration, its isotopic composition, nitrogen concentration and terrigenous biomarkers in circum-Arctic marine sediments. This is an original and useful compilation, as it makes data published in different sources easily accessible, previously unpublished data available and even complements existing data with gap-filling measurements and spatial interpolation using GIS. Such data is needed for integrative and large-scale assessment of biogeochemical cycles, especially in regions as sensitive to changing climate as the Arctic. First insights are deduced and visualized, identifying spatial variation in organic carbon concentration and sources, which gives the reader an impression of how to use and interpret this data. The provided dataset is well organized, curated, (re)traceable and citable via DOI, despite a few apparent transcription errors (which are common in these types of manual compilations) and minor comments on comprehensibility (see Dataset comments). The paper itself describes data derivation and classification in a reasonable way and with sufficient detail, although (apart from minor comments) the reader-friendliness of the structure, consistency and conciseness could be improved.*

> We are pleased about this overall positive reception of the CASCADE effort and thank referee #1 for the constructive feedback to the manuscript and for suggesting possible improvements of the clarity and comprehensibility of the text. The comments and suggestions detailed below have improved the clarity and quality of the paper.

**Specific Comments**

**Abstract**

*The Abstract includes all relevant information, but could be more concise. For instance, the beginning (L.30 – 34) can be reformulated more concise and your interpretation presented in L. 46 – 50 may be explained more briefly. Possibly, try to avoid too many adjectives/adverbs and embedded subordinate clauses.*

> We thank the reviewer for this comment and agree that the abstract ought to be written more concisely. In the revised manuscript we have shortened the abstract, e.g. by condensing the text

between lines 30-33 and between lines 46-48. We have also critically re-evaluated the full manuscript text with an eye to improve and shorten sentence structures.

*Introduction*

*1.) In L. 58: '[..] large input of terrestrial organic matter from its large rivers and from coastal erosion, making it both a valuable receptor system for studying large-scale terrestrial carbon remobilization and marine biogeochemistry', consider substitution of one or two 'large'. There is also a semantic error: '[. . .] making it both a valuable receptor system [. . .] and marine biogeochemistry.' You may try to relocate 'both'.*

Thank you for this comment. We have now rewritten this sentence as follows:

*"The Arctic Ocean receives large input of terrestrial organic matter from rivers and coastal erosion, making it a valuable receptor system for studying both large-scale terrestrial carbon remobilization and marine biogeochemistry. "* line 56-58

*2.) References are needed to support your statement, that warming accelerates coastal erosion and river runoff in L. 60 - 61 (although this is senseful of course). Similarly, citations should be added in L. 69-70 (arctic warming as a tipping point in the climate system) and L. 75 – 79 (global and arctic shelf area portions). In L. 89 – 90, where you mention 'Key progress', the (or some) relevant articles should be cited. If that refers to the references in the following explanatory sentences, please make that more clear, e.g. by inserting ':' .*

We agree with the reviewer that these statements should be supported by the key references. These are now cited as follows:

Line 58-60: *"Rising temperatures cause multiple changes to the Arctic, including reduced sea-ice cover, accelerated erosion of ice-rich permafrost shorelines and enhanced river runoff, which changes the input of terrestrial organic matter to the Arctic Ocean (AMAP, 2017)."*

Line 66-68: *"Couplings between the large permafrost-carbon pools and amplified climate warming in the Arctic represent a potential "tipping point" in the climate system (Lenton, 2012)."*

Line 72-77: *"Continental shelves cover less than 10% of the global ocean area but account for the largest part of OC accumulation in marine sediments and thereby provide an excellent archive for both terrestrial carbon input and marine productivity (Hedges et al., 1997). The Arctic Ocean is semi-enclosed and dominated by its extensive shelves, including the World's largest continental shelf system, the East Siberian Arctic Shelf (ESAS; the Laptev, East Siberian and Russian part of the Chukchi Sea). This further accentuates the particular importance of shelf sediments for carbon cycling in the Arctic (Stein et al., 2004; Vetrov and Romankevich, 2004)."*

The sentence in line 86-88 refers to the studies listed in the following sentences. The sentences was revised to: *"Substantial progress was made by individual and region-specific studies since then; with key advances in isotope and organic geochemistry that expand the variety of biogeochemical proxies to trace both sources and organic matter degradation. […]"*

*3.) In L. 108 – 109, you state that there is an initial focus on terrigenous organic matter, but not why. Is that because of data availability or because of the database applications you had in mind? Please give the reasoning here.*

It is correct that the primary motivation for this database is to use sediments as archive for circum-Arctic terrestrial organic matter input. However, while CASCADE is facilitating the

study of terrestrial carbon release at a large-scale, it is also intended to stimulate studies beyond this research focus. We encourage studies of biogeochemical cycling but also of organic contaminants in the Arctic Ocean. The revised manuscript now contains a line that states:

*"The Circum-Arctic Sediment CArbon DatabasE (CASCADE) builds on previously-published and unpublished collections holding information on OC and total N (TN) concentrations, as well as OC isotopes ($\delta^{13}$C-OC, $\Delta^{14}$C-OC) in sediments of all continental shelves and the deep central basins of the Arctic Ocean. Furthermore, CASCADE contains molecular data with an initial focus on terrestrial biomarkers (i.e., high molecular weight - HMW n-alkanes, n-alkanoic acids, lignin phenols) to facilitate studies of terrestrial OC remobilization."* line 102-107

*Data collection and methods*

*1.) Section 2.2 Georeferencing and sampling is a bit confusing (e.g., what do you mean by 'core part'?), which is clarified later in section 2.4 Database parameters. So maybe consider explaining the parameters and structure before you give details on how these parameters are acquired.*

We thank the reviewer for this comment and agree that the '*core part*' wording is unclear. Section 2.2 aims to provide the geographical framework, which is in 2.3 distinguished between different depth categories. All carbon parameters are explained in 2.4. We believe that this order makes sense and allows readers to follow the structure of CASCADE. To improve the clarity, we thoroughly revised and re-ordered section 2.2 Furthermore, the sentence in line 141 now clarifies that "*The collection of data from oceanographic stations is the main part of CASCADE and is organized in a table format that contains columns* […]".

*2.) L. 151 states that you used '[. . .] the year of the earliest published record [. . .]' when the sampling date was not available. This needs to be visible in the data tables, for full comprehensibility and because publication and sampling year may differ in fact. Is more detailed sampling time information available? Seasonality may play a role in surface sediments.*

We agree with the reviewer that it should be clear if the sampling date was available or not. However, with our definition of surface sediments (to max. 5 cm depth) we cannot resolve seasonal cycles. To improve clarity of the availability of the sampling year we removed the year for cases where no year was reported. In the revised manuscript it now states "*For samples where the sampling year was unknown, users may use the year of publication instead*." (line 149-150).

*3.) In section 2.3 you could possibly provide the number of sediment cores available for each of the scales (Centennial, Millennial and Glacial cycles scale) and provide a reasoning for this separation (i.e., how did you choose length range and corresponding time-scale?)*

The distinction between these three time scales is motivated by environmental and paleo-climate studies that often distinguish between these scales, and represent very different types of research questions that may be addressed using such data. The revised manuscript now includes a line about the motivation for the distinction:

*"Types of sediment cores are distinguished in CASCADE such that different biogeochemical processes, acting on three depositional time scales, may be addressed. The three time scales are:"* line 158-160

The number of sediment cores in these three categories are now detailed in lines 308-309:

*"In addition to surface sediments, a total number of 326 sediment cores (79 centennial, 229 millennial, and 18 glacial cycle scale cores) is included in the first version of CASCADE."*

*4.) In L. 206 – 209, it is stated that for some samples, variables are from different references. Is that exactly the same sample (or a split)? How do you assure this? Is it explicitly stated by the references?*

For these cases of multiple references at one station we made sure that the measurements were made using the same sample material. However, we cannot resolve any possible subsampling during or after the cruise from the published literature. In the revised manuscript this is now clarified by changing the line in 206-209 "*This facilitates to register multiple measurements based on the same or split sediment sample material for individual oceanographic stations.*".

*5.) The detailed description of the included parameters in section 2.4 Database parameters is (partially) repeated in section 3.1 Data set inventory, where you basically make a similar detailed description, but with numbers of samples.*

We agree that there strictly speaking is some redundancy between these two sections. The structure is that Section 3.1 presents the resulting number of observations based on the criteria elaborated in 2.4. However, we see no simple way of presenting these important numbers about the database inventory without connecting to the parameters at this level of detail.

*6.) Section 2.6 Data source and quality. In this section you describe 'The quality criteria for data to be included [. . .]'. These appear well chosen at the first glance, but later in the evaluation of the data quality, it becomes obvious that not all data fulfill these criteria (e.g. 'For $\delta^{13}$C-OC, in 66 % of the cases IRMS coupled to EA was given [. . .]' (L. 333 -334) and the rest?). Therefore, you may not call these selection criteria, but quality assurance criteria or similar. At least stay consistent with this.*

Thank you for making this very important point! We agree with the reviewer that these criteria were not used to exclude data from the database and find that the term *quality criteria* may be misleading here. While we have in fact not used the term *selection criteria* in the text, we chose to recast the wording to *quality assurance criteria*, and now clarify that these criteria are not used to exclude data from CASCADE but to provide maximum quality assurance and transparency to the end user:

"*To facilitate quality assurance criteria by the end users the database also records metadata (e.g., sampling technique in the field, sample storage) and quality data when available. The quality assurance information for data in CASCADE are:*" line 217-219

*7.) In section 2.7.1, L. 259 – 260, you give the precision of the gap-filling measurements, but neither mention how this was acquired (multiple measurements?), nor include a statement regarding accuracy.*

Thanks for drawing our attention to this. We have clarified regarding these precisions by adding lines 255-257 "*The measurements at Uppsala University had a precision of on average ±1.9‰ while the precision at ETH Zurich was on average ±1.1‰ (based on $^{14}$C counting statistics).*"

*8.) The constants used for conversion of $^{14}$C data should be referenced (L. 282 – 284).*

We agree with the reviewer and have added the respective references (Stuiver and Polach, 1977) in the revised manuscript.

*Results and Discussion*

*1.) Section 3.1: Partial repetition of section 2.4. See comments in previous section.*

We addressed this point above.

*2.) Section 3.3: Does your remark in L. 332-334, '[. . .] CASCADE provided detailed information [. . .] or cited references or cruises that contained this information', mean that it may be up to the user to look up methods for some samples in the original reference? If yes, why didn't you include this information for all samples then? If no, please try to reformulate to avoid confusion.*

No, this is a misunderstanding. We have included the quality assurance information in CASCADE as described in the manuscript in line 335. To make this point clearer we have revised this sentence accordingly: "*Based on the quality assurance data available, CASCADE provides detailed information about the techniques involved in analyzing OC concentrations, isotopes and biomarkers.*" line 330-331

*3.) Which external data sources are available that can be interoperated with CASCADE? It may be instructive to compare marine sediment data to river input, water column chemistry, marine production, coastal and soil erosion or the non-organic carbon part of marine sediment. At least for some of these parameters, databases exist (e.g. GLORICH https://doi.org/10.1594/PANGAEA.902360 or db seadbeds https://instaar.colorado.edu/~jenkinsc/dbseabed/). You may stimulate users by providing some advice here.*

We thank the reviewer for this suggestion to point to other existing databases. We believe that CASCADE provides ground for a large variety of applications and study directions. The sheer amount of other dataset would make any selection arbitrary, which is why we refrain from doing so.

*4.) You may possibly want to insert a disclaimer in the end ([https://www.earth-system-sciencedata.net/submission.html#manuscriptcomposition](https://www.earth-system-sciencedata.net/submission.html#manuscriptcomposition)).*

Thank you for this suggestion. We have consulted the Editorial office to ask about the need and purpose of a disclaimer section. It is reserved for cases where authors need to clarify legal aspects of their work. Based on this we currently do not see the need to include such a section.

*Dataset*

*1.) CASCADE is very well structured, comprehensible and user friendly. However, an indication for what type of time-reference is given should accompany that valuable information. In the current version, a user cannot know if a sampling time refers to the real sampling year or to the earliest publication year from the same sample/core, which may differ by some years.*

We have addressed this point above. In the updated database, the sampling year is only given when the year is known. In case no sampling year was reported the entry is kept blank.

*2.) The dataset has proven robust against detailed inspections and logical tests. In spite, there is a little transcription (?) error, identified by the histogram of '% OC' in 'CASCADEcoresv1', indicating an impossible organic carbon concentration of ~120 %, which probably propagated into the following calculations (C/N ratios and normalization to organic carbon concentration). Please check your calculations, possible errors in unit conversion and transcription errors (digit typo?) and correct this.*

We greatly appreciate the reviewer for pointing out an error in one of the data sheets in CASCADE. The value for total OC in row ID 6045 in data table 'CASCADEcores_v1.0' and in row ID 5555 in table 'CASCADEcorescale2_v1.0' (133 wt% OC), imported from another data source, is indeed incorrect. The value will be removed from our revised version of CASCADE.

*3.) A table translating abbreviations used in the database (e.g. IRMS) and units should accompany the data for clarity and convenience, despite this information is included in the article.*

We agree with the reviewer that such information would ideally be included in all CASCADE data tables but this is, unfortunately, not possible for some of the data formats (.txt). We have now added a list of abbreviations in the Excel-files in the updated version of CASCADE, which is likely the format that will be employed by most users. To the other file formats we added "README" text files with more information about the units.

*4.) If possible, please provide the interpolated grid also as datafile to complement the rather impractical tiff-file.*

This is a very good suggestion. The interpolated grids are now also available as ASCII txt files in addition to the tiff images.

**Referee comment 2**

*General Comments:*

*Martens et al. present a new, openly accessible database called "CASCADE" compiling various data from surface sediments and sediment cores from the Arctic Ocean. Specifically, the database contains data on TN and TOC contents, carbon isotopic compositions of bulk organic matter ($\delta^{13}C$ and $\Delta^{14}C$) and concentrations of terrigenous biomarkers (n-alkanes, n-alkanoic acids and lignin phenols). The authors combine existing data from different databases and even make previously inaccessible data available for the community. They also generate new data to fill regional gaps. At the end of the paper, they apply CASCADE to interpolate carbon concentrations and bulk $^{13}C$ and bulk $^{14}C$ data over the Arctic Ocean and discuss regional differences.*

*The paper and the database are clearly arranged and comprehensible. The data is easy to access as it is provided in common formats including text, excel and shape files. The quality standards and methods applied are state of the art and robust.*

*The database is a valuable piece of work and will be vital for Arctic research as it allows to comprehensively study biogeochemical processes and carbon cycling in the Arctic Ocean and how these processes are coupled to changes in the adjacent continents (e.g. permafrost thawing). Understanding these processes is crucial in a region expected to change drastically in the course of global warming. Altogether the database and paper stand already very well by themselves and do not need major editing. The paper can be published soon after a few small issues are addressed.*

> We are pleased by the overall positive reception of both the database and the manuscript, and we thank the reviewer for the valuable and constructive comments that contributed to better quality and clarity of the manuscript.

*1) Overall the paper is well structured and easy to understand. However, it contains some rather long sentences that should be shortened to facilitate reading. Moreover, I found a few instances where the word order should be rearranged (see Specific Comments below)*

> We appreciate the reviewer for making this comment. We have tried to shorten sentences throughout the manuscript, e.g. long sentences in lines 58-62 or 299-301 were split in two sentences to improve the comprehensibility and flow to the reader. We have also edited the manuscript text for straight word order.

*2) Although the authors describe the file contents on the website and in the paper, I find it unhandy that the data files do not contain units in the column headers. The authors should consider to add the units to the files to make the work with the data even more comfortable for the user.*

> This is a good and valid point. In the updated CASCADE version we have now added this information in an abbreviated form to the column headers of the data tables. We have also added an overview table of all parameters and abbreviations to the Excel spreadsheets and as "README" text files, as this was addressed by the other reviewer.

*Specific comments:*

*Line 43: change "shelves" to "Shelves"*

> The word was removed during the revisions.

*Line 111: I think "$^{13}C/^{14}C$-isotope data" actually means $\delta^{13}C$ and $\Delta^{14}C$ data. This should be clearly stated as the term may be misunderstood as the isotopic ration between $^{13}C$ and $^{14}C$.*

We agree that this should be consistent throughout the manuscript. The revised manuscript now consistently uses $\delta^{13}C$ and $\Delta^{14}C$.

*Line 158: change to: "Here, surface sediments are defined as. . ."*

The sentence was changed according to the reviewer suggestion. Line 154

*Section 2.7.1: Does the gap filling concern surface sediments and cores?*

No, the gap filling concerns only surface sediments. This is now clarified in the revised manuscript:

"*Gap filling was performed in surface sediments of regions with particularly poor data density.*" line 241

*Line 366: change to: towards remobilization in both, the current and over earlier. . ..*

The sentence was changed according to the suggestion. line 363

**References**

AMAP, 2017. *Snow, Water, Ice and Permafrost in the Arctic (SWIPA) 2017. Arctic Monitoring and Assessment Programme (AMAP), Oslo, Norway.*

Hedges, J.I., Keil, R.G., Benner, R., 1997. *What happens to terrestrial organic matter in the ocean?, in: Organic Geochemistry. Pergamon, pp. 195–212. https://doi.org/10.1016/S0146-6380(97)00066-1*

Lenton, T.M., 2012. *Arctic climate tipping points. Ambio. https://doi.org/10.1007/s13280-011-0221-x*

Stein, R., Macdonald, R.W., Naidu, A.S., Yunker, M.B., Gobeil, C., Cooper, L.W., Grebmeier, J.M., Whitledge, T.E., Hameedi, M.J., Petrova, V.I., Batova, G.I., Zinchenko, A.G., Kursheva, A. V, Narkevskiy, E. V Fahl, K., Vetrov, A., Romankevich, E.A., Birgel, D., Schubert, C., Harvey, H.R., Weiel, D., 2004. *Organic Carbon in Arctic Ocean Sediments: Sources, Variability, Burial, and Paleoenvironmental Significance, in: Stein, R., MacDonald, R.W. (Eds.), The Organic Carbon Cycle in the Arctic Ocean. Springer Berlin Heidelberg, Berlin, Heidelberg, pp. 169–314. https://doi.org/10.1007/978-3-642-18912-8_7*

Stuiver, M., Polach, H.A., 1977. *Discussion Reporting of 14C Data. Radiocarbon 19, 355–363. https://doi.org/10.1017/S0033822200003672*

Vetrov, A.A., Romankevich, E.A., 2004. *Carbon Cycle in the Russian Arctic Seas. Springer Berlin Heidelberg, Berlin, Heidelberg. https://doi.org/10.1007/978-3-662-06208-1_7*